# Silencing of LLGL2 Suppresses the Estradiol-Induced BPH-1 Cell Proliferation through the Regulation of Autophagy

**DOI:** 10.3390/biomedicines10081981

**Published:** 2022-08-16

**Authors:** Kyung-Hyun Kim, Geum-Lan Hong, Yae-Ji Kim, Hui-Ju Lee, Ju-Young Jung

**Affiliations:** Department of Veterinary Medicine & Institute of Veterinary Science, Chungnam National University, 99, Daehak-ro, Yuseong-gu, Daejeon 34134, Korea

**Keywords:** LLGL2, benign prostatic hyperplasia, autophagosome formation, proliferation

## Abstract

Lethal giant larvae (Lgl) is an apical-basal polarity gene first identified in *Drosophila*. LLGL2 is one of the mammalian homologs of Lgl. However, little is known about its function in the prostate. In this study, to explore the new role of LLGL2 in the prostate, we examined the proliferative activity of a BPH-1 cell line, a well-established model for the human prostate biology of benign prostatic hyperplasia (BPH). The expression of LLGL2 was dose-dependently increased in BPH-1 cells after treatment with 17β-estradiol (E2). Additionally, E2 treatment increased the proliferation of the BPH-1 cells. However, the knockdown of LLGL2 with siRNA significantly suppressed the proliferation of the E2-treated BPH-1 cells. Moreover, si-*llgl2* treatment up-regulated the expression of LC-3B, ATG7, and p-beclin, which are known to play a pivotal role in autophagosome formation in E2-treated BPH-1 cells. Overexpression of LLGL2 was able to further prove these findings by showing the opposite results from the knockdown of LLGL2 in E2-treated BPH-1 cells. Collectively, our results suggest that LLGL2 is closely involved in the proliferation of prostate cells by regulating autophagosome formation. These results provide a better understanding of the mechanism involved in the effect of LLGL2 on prostate cell proliferation. LLGL2 might serve as a potential target in the diagnosis and/or treatment of human BPH.

## 1. Introduction

Benign prostatic hyperplasia (BPH) is characterized by non-cancerous prostate gland enlargement. BPH occurs commonly in aging men. Increased cell numbers in the epithelial and stromal cells of the prostate are among the main etiologies of BPH [1]. Traditionally, the development of BPH requires the presence of testicular androgen as the prostate is a hormone-response organ. In prostate cells, androgen is transformed into dihydrotestosterone (DHT) by type II 5α-reductase. These signaling pathways have been considered to be the main signaling pathways for BPH development [2]. Emerging evidence has shown that BPH regulated by DHT is far from the clinical consequences of aging-related BPH etiology. In humans, as serum androgens decline with advancing age, serum estradiol (E2) levels remain relatively constant. However, the net effect is an increased serum E2 to testosterone (T) ratio [3]. Moreover, in situ local production of E2 has been implicated in prostatic hyperplasia. It has been shown that loss of aromatase expression causes decreased estrogen-induced prostate proliferation [3,4]. In this regard, a high serum E2 to T ratio in the prostate may contribute to the progression of BPH processes.

*Lethal giant larvae* (*Lgl*) is an apical-basal polarity gene first identified in *Drosophila* [5]. The maintenance of cell polarity is the most essential property of normal cells. It is crucial to regulate various biological processes, such as differentiation, proliferation, adhesion, migration, and tumor formation [6,7]. In *Drosophila*, *Lgl* functions as a tumor suppressor, controlling the self-renewal and differentiation of progenitor cells [8,9]. There are two orthologs of *Lgl* in mammalian genomes: *Llgl1* and *Llgl2*. However, little is known about their functions in vivo. Several studies have shown that LLGL1 is a tumor suppressor in various human cancers [10,11,12]. LLGL2 is also associated with cancer progression. It is vital for asymmetric cell division and migration [13,14]. LLGL2 can suppress Snail-induced epithelial-mesenchymal transition (EMT) as a tumor suppressor, preventing the dissemination of breast cancer [15]. Nevertheless, Saito et al. have recently reported that LLGL2 functions as a tumor promoter, not a tumor suppressor, by assisting cancer cells in overcoming nutrition stress in human ER+ breast cancer [16]. Leng et al. have also shown that LLGL2 can facilitate the proliferation and migration as well as the metastasis in hepatocellular carcinoma [17]. However, the biological implications of LLGL2 in the prostate, especially BPH, remain unknown.

Thus, this study investigated the expression of LLGL2 in BPH. Furthermore, we examined the role of LLGL2 in the E2-mediated proliferation of BPH-1 cells, which in turn could influence autophagic pathways.

## 2. Materials and Methods

### 2.1. Animal Experiments and Immunohistochemical Staining

The experimental protocols using animals were approved by the International Animal Ethics Committee at Chungnam National University (202009A-CNU-147). Male Sprague–Dawley (SD) rats (n = 30, six-weeks-old) were purchased from Orient Bio (Gyeonggi-do, Korea) and acclimated for one week. These SD rats were placed in a specific-pathogen-free (SPF) room under controlled temperature (22 ± 2 °C), humidity (55 ± 5%), and photoperiod (12 h light/12 h dark cycle). All the animals were fed water and standard chow ad libitum. The rats were given subcutaneous (S.C.) injections of T (5 mg/kg dissolved in corn oil) for four weeks to induce BPH. Following the final injection, the rats were fasted overnight and euthanized by CO_2_ asphyxiation in a euthanasia apparatus. Their prostate tissues were excised, fixed immediately in a 10% buffered formalin phosphate solution, embedded in paraffin, and cut into 5 μm sections. These sections were subjected to deparaffinization and hydration. The prostatic antigens were retrieved with Tris-EDTA solution through heating under steam for 10 min. The sections were treated with 0.3% (*v*/*v*) hydrogen peroxidase in methanol for 15 min to block endogenous peroxidase activity. They were blocked with 1.5% normal goat serum for 1 h. These blocked specimens were incubated with antibodies against LLGL2 (1:200, Santa Cruz, Dallas, TX, USA) in a humidified chamber overnight at 4 °C. After washing with PBS, the sections were incubated with horseradish peroxidase-conjugated anti-mouse IgG antibody (1:200; AbFrontier, Seoul, Korea) in a humidified chamber for 1 h at room temperature (RT). A 3,3’-diaminobenzidine substrate kit (Vector Labs, Burlingame, CA, USA) was used to visualize signals under a microscope. The slides were examined using an Eclipse 80i microscope (Nikon, Tokyo, Japan) and evaluated. Ten randomly selected fields were evaluated for each slide.

### 2.2. Cell Culture and Reagents

Human BPH-derived prostate epithelial cell lines (BPH-1) were purchased from Merck Millipore (Merck, Kenilworth, NJ, USA). The cell line was cultured in RPMI 1650 (Gibco, Grand Island, NY, USA) medium with 1% penicillin/streptomycin (Gibco) and 10% fetal bovine serum (FBS, Gibco) at 37 °C in a 5% CO_2_ atmosphere. The cells were seeded into 6-well plates at a density of 3.0 × 10^5^ cells/well in 2 mL of medium. E2 was purchased from Sigma Aldrich (St. Louis, MO, USA).

### 2.3. siRNA and Plasmid Transfection

The BPH-1 cells were transfected with LLGL2 siRNAs (Santa Cruz) using Lipofectamine™ RNAiMAX Transfection Reagent (Thermo Fisher Scientific, Waltham, MA, USA) for 48 h according to the manufacturer’s instructions. Firstly, 30 p moles of siRNA were diluted in 150 μL of OPTI-MEM (Gibco) medium, and 9 μL of Lipofectamine™ RNAiMAX Reagent was diluted in 150 μL of OPTI-MEM medium. Diluted LLGL2 siRNAs were then mixed with Lipofectamine™ RNAi MAX Reagent (1:1 ratio) and incubated at RT for 5 min. During the incubation time, 2 mL of serum-containing medium was removed and a charcoal-dextran-treated FBS (CDS, Gibco)-containing medium was added. Finally, the transfection mix was dropped into cells in a CDS-containing medium. The efficiency of the LLGL2 knockdown was examined by Western blot. To overexpress the LLGL2, the *LLGL2* gene was sub-cloned into pEXPR-IBA105 vectors to obtain *LLGL2* plasmids. The BPH-1 cells were transfected with the *LLGL2* plasmids with Lipofectamine 2000 Reagent (Thermo Fisher Scientific) according to the manufacturer’s instructions. The efficiency of the LLGL2 overexpression was then examined by Western blot.

### 2.4. Cell Viability Assay

The BPH-1 cells were seeded in a 96-well plate at a density of 1 × 10^4^ cells/well and incubated in a 5% CO_2_ atmosphere at 37 °C for 24 h. The cells were treated with LLGL2 siRNAs for 48 h. Next, the cells were treated with E2 (0, 10, and 100 nM) for 24 h. The cell proliferation was examined with the EZ-Cytox Cell Viability Assay Kit (Biomax, Seoul, Korea) according to the manufacturer’s instructions. Briefly, 10 μL of EZ-Cytox working solution per 100 μL of the medium was added to the 96-well plate, and the cells were incubated at 37 °C for 1 h. Absorbance was measured with a microplate reader (BIO-TEK, Senergy HT). The cell viability was calculated as 100% × (OD_450 nm_ of *S. mirabilis* extract group/OD_450 nm_ of the control group).

### 2.5. Observation of Cell Confluency

The BPH-1 cells were seeded in a 6-well plate at a density of 1 × 10^5^ cells/well and incubated in a 5% CO_2_ atmosphere at 37 °C for 24 h. The BPH-1 cells were treated with LLGL2 siRNAs for 48 h and were treated with E2 (0 and 100 nM) for 24 h. The cell morphology was observed with a phase-contrast microscope (Olympus, Tokyo, Japan). Photographs were taken at a magnification of 100×.

### 2.6. Flow Cytometry

The BPH-1 cells were seeded in a 6-well plate at a density of 1 × 10^5^ cells/well and incubated in a 5% CO_2_ atmosphere at 37 °C for 24 h. The BPH-1 cells were treated with LLGL2 siRNAs for 48 h and were treated with E2 (0 and 100 nM) for 24 h. The cells were washed with PBS and were fixed with ice-cooled 70% ethanol at 4 °C for 1 h. The BPH-1 cells were stained with propidium iodide (PI, Invitrogen) and analyzed by flow cytometry (BD Bioscience, FranklinLakes, NJ, USA). The cell proliferation was expressed as the percentage of S-phase cells.

### 2.7. Western Blot Analysis

The cells were lysed in a Radio-Immunoprecipitation Assay (RIPA) buffer (Cell Signaling Technology, Danvers, MA, USA) in the presence of a protease inhibitor cocktail (Roche, Mannheim, Germany) for 15 min on ice and centrifuged at 12,000 rpm for 15 min at 4 °C. The supernatant was collected and used for Western blotting. Protein samples (10 μg/well) were separated by 6–12% sodium dodecyl sulfate polyacrylamide gel electrophoresis (SDS-PAGE). Resolved proteins were transferred to polyvinylidene fluoride membranes with a wet transfer system (Bio-Rad, Hercules, CA, USA). The membranes were blocked with 5% (*w*/*v*) skim milk in 1× phosphate buffered saline (PBS) containing 0.1% Tween-20 (PBS-T). These membranes were then incubated with antibodies against LLGL2, ATG7 (1:1000, Santa Cruz), LC3 (1:1000, Sigma-Aldrich), CyclinD1 (1:1000, Cell Signaling Technology, Danvers, MA, USA), p-beclin, or β-actin (Abcam, Cambridge, UK) at 4 °C overnight. After washing three times with PBS-T, the membranes were incubated with a horseradish peroxidase-conjugated IgG (anti-rabbit or anti-mouse) secondary antibody (1:5000, AbFrontior, Seoul, Korea). Each protein expression was detected using an enhanced chemiluminescence detection kit (Thermo Fisher Scientific). Signal intensities were quantified using Image Lab (Bio-Rad).

### 2.8. Immunofluorescence

The BPH-1 cells were fixed in 4% paraformaldehyde for 15 min, washed with PBS three times, and blocked with 3% bovine serum albumin for 30 min. These cells were then incubated with anti-LC3 (1:500, Sigma-Aldrich) at 4 °C overnight. The cells were washed with PBS and incubated with Goat Anti-Rabbit IgG (Alexa Fluor^®^ 488) conjugated secondary antibodies (Abcam) at RT for 1 h. The nucleus was counterstained with 4′6-diamino-2-phenylinodole (DAPI, Vector Labs). Zeiss LSM 880 with Airyscan confocal microscopy (Carl Zeiss, Jena, Germany) was performed at 488 and 592 nm wavelengths. Images were captured with ZEN software.

### 2.9. Detection of Autophagy Flux

The formation of autophagosome and autolysosome in the BPH-1 cells was detected using a Premo™ Autophagy Tandem Sensor RFP-GFP-LC3B Kit (Thermo Fischer Scientific, Grand Island, NY, USA) following the manufacturer’s instructions. The RFP-GFP-LC3B sensor could detect LC3B positive, neutral pH autophagosomes in green fluorescence (GFP) and LC3B positive, acidic pH autophagolysosme in red fluorescence (RFP). The BPH-1 cells were grown on coverslips and incubated with 6 μL of BacMam Reagents containing RFP-GFP-LC3B overnight. These cells were then treated with LLGL2 siRNAs alone for 48 h. The cells were washed with PBS three times. The nucleus was counterstained with DAPI (Vector Labs). Zeiss LSM 880 with Airyscan confocal microscopy (Zeiss) was performed at wavelengths of 488 and 592 nm. Images were captured with ZEN software.

### 2.10. Statistical Analysis

The data are expressed as mean ± standard deviation (SD). All statistical analyses were performed using one-way ANOVA with a Tukey’s multiple comparison test. *p*-values of less than 0.05 were considered statistically significant.

## 3. Results

### 3.1. LLGL-2 Expression in Testosterone-Induced Rat BPH Animal Model

To investigate the expression of LLGL2 in BPH in vivo, we used normal rat prostate and testosterone-induced rat prostate for IHC and Western blotting. In the testosterone-induced BPH tissues, LLGL2 was especially overexpressed in the epithelial region of the prostate (Figure 1B,D). However, the normal rat prostate barely had LLGL2 expression (Figure 1A,C). To confirm this finding, the LLGL2 expression level in prostate tissue was analyzed by Western blotting. Consistent with the results of the IHC analysis, the expression of LLGL2 was increased in testosterone-induced BPH tissues compared to that in the normal prostate tissue (Figure 1E,F).

### 3.2. LLGL2 Expression in Estradiol-Induced BPH-1 Cell Proliferation

During the progression of BPH, the hormone is a pivotal regulatory factor for the proliferation of the prostate. Testosterone is one of the most potent hormones for prostatic proliferation. Recently, it has been shown that E2 can also affect prostatic cell proliferation [18]. In this regard, we treated the BPH-1 cells with E2 to analyze its effect on cell proliferation. Upon E2 treatment, the cell proliferation was observed by MTT assay (Figure 2A). Notably, the expression of LLGL2 was up-regulated by the E2 treatment in the BPH-1 cells. Additionally, the expression of Cyclin D1 was increased in the E2-treated BPH-1 cells (Figure 2B). The knockdown of LLGL2 using siRNA significantly reduced the expression of LLGL2 in the BPH-1 cells. The expression of Cyclin D1 was also decreased by si-*llgl2* in the E2-treated BPH-1 cells (Figure 3A). Morphologically, the number of adherent cells was increased after the E2 treatment, whereas the knockdown of LLGL2 reduced the number of adherent cells (Figure 3B). Moreover, the viability of the BPH-1 cells was significantly decreased after si-*llgl2* treatment in the E2-treated BPH-1 cells compared to that in the E2-treated BPH-1 cells with si-*cont* treatment (Figure 3C). Cell cycle analysis also showed that the E2-treated BPH-1 cells with si-*cont* elevated in the S-phase, while the E2-treated BPH-1 cells with si-*llgl2* showed a decrease in the S-phase (Figure 3D,E). Taken together, these results demonstrate that the silencing of LLGL2 expression could reduce the E2-induced proliferation of BPH-1 cells.

### 3.3. Inhibition of LLGL2 Expression Induces Autophagy

Several studies have shown a close correlation between cell cycle responses and autophagy [19,20]. Next, we examined whether the silencing of LLGL2 expression could regulate autophagy in E2-induced BPH-1 cells. The silencing of LLGL2 expression remarkably induced LC3-II expression and increased LC3 puncta in the E2-treated BPH-1 cells (Figure 4A,B). The expression levels of ATG7 and p-beclin, closely related to autophagosome formation, were significantly increased in the si-*llgl2*-E2-treated BPH-1 cells. The silencing of LLGL2 resulted in a decreased level of p62 expression (Figure 4A). To visualize the effect of silencing the LLGL2 gene in the E2-treated BPH-1 cells, autophagosomes were stained with a specific tandem RFP-GFP-tagged LC3. Consistent with the protein expression results of LC3 and LC3 puncta, the yellow fluorescent autophagosomes were increased after the LLGL2 knockdown in the E2-treated BPH-1 cells (Figure 4C). These results indicate that the silencing of LLGL2 expression can induce autophagosome formation in E2-treated BPH-1 cells.

### 3.4. Overexpression of LLGL2 Promotes the Proliferation of BPH-1 Cells

To further examine the effects of LLGL2 on the proliferation and autophagy of E2-treated BPH-1 cells, the overexpression (O/E) LLGL2 vector was used to transfect E2-treated BPH cells. The expression of Cyclin D1 was increased in the LLGL2 O/E vector-treated group. Additionally, there was no significant change in the expression of LC3-II due to the overexpression of LLGL2 (Figure 5). Collectively, these findings suggest that the overexpression of LLGL2 shows the opposite results from the knockdown of LLGL2 in E2-treated BPH-1 cells.

## 4. Discussion

In this study, we first assessed the expression levels of LLGL2 in the prostate, especially in the BPH. We found that the expression of LLGL2 was increased in testosterone-induced rat BPH in vivo. In the BPH-1 cells, E2 treatment affected the cell viability and upregulated the expression of LLGL2. The silencing of LLGL2 expression inhibited cell proliferation and induced autophagosome formation in the E2-treated BPH cells. Moreover, the overexpression of LLGL2 stimulated the proliferation of E2-treated BPH cells. Our results are noteworthy because the role of LLGL2 in proliferation of the prostate is currently unknown.

McNeal has proposed one prominent theory of BPH pathogenesis, which is that hyperplasia of the prostate is induced by DHT, which is an essential metabolite of T and the most potent androgen in men [21]. However, various studies have shown that androgens are permissive but insufficient for the induction and maintenance of BPH. Androgens might not influence prostate growth because the supplementation of men with androgens does not appear to increase the incident risk of BPH or to lower urinary tract symptoms (LUTS) [22]. Furthermore, BPH prevalence increases with age, while levels of serum androgens decline. On the other hand, the serum levels of E2 remain relatively constant, although the net effect is an increased ratio of serum E2 to T. E2 is considered the most potent estrogen in men. It is essential for various physiologic processes, including bone maturation and mineralization, peak bone mass, and skin and lipid metabolism [23]. Recently, Saito et al. have reported that E2 in ER+ breast cancer can control the regulation of LLGL2. We focused on these mechanisms of the action of E2 regarding cell proliferation and applied them to BPH-1 prostate epithelial cell lines. Our results showed that E2 treatment induced cell proliferation and regulated the S-phase of BPH-1 cells. Moreover, E2 treatment increased the expression of LLGL2 in BPH-1 cells. Consistent with the in vitro results, the expression of LLGL2 was also upregulated in the testosterone-induced rat BPH prostate. These results demonstrate that LLGL2 is closely related to prostate cell proliferation.

Autophagy is a highly conserved evolutionary and complex cellular process in eukaryotic cells. Cytoplasmic long half-life proteins and organelles are sequestered within autophagosomes and delivered to lysosomes for degradation and recycling [24]. Autophagy is a critical regulatory mechanism for cell proliferation and death. Several reports have revealed that autophagy is involved in BPH development [25,26]. The inhibition of androgen can induce autophagy in benign prostatic epithelial cells [27]. Jiang et al. have shown that ATG9A is upregulated after long-term 5α-reductase treatment in BPH progression [28]. In this regard, autophagy is helpful against the proliferation of the prostate by protecting intracellular homeostasis [19,29]. In this study, silencing LLGL2 expression upon E2 treatment showed enhanced autophagy through the upregulation of LC3-II, ATG-7, and p-beclin and the downregulation of p62. LC3-II, ATG-7, p-beclin, and p62 are known to be key regulatory molecules for autophagosome formation [24]. These results suggest that the silencing of LLGL2 expression might regulate the autophagosome formation in E2-treated BPH-1 cells. Moreover, the overexpression of LLGL2 was able to further prove these findings by showing the opposite results from the knockdown of LLGL2 in E2-treated BPH-1 cells. The present study revealed that silencing LLGL2 could suppress the proliferation of BPH-1 cells.

Overall, we suggested the role of LLGL2 in the proliferation of the prostate by the regulation of autophagy processes. This is the first attempt to study the role of LLGL2 in the prostate. These results provide a better understanding of the mechanisms involved in the role of LLGL2 in prostate cell proliferation. LLGL2 might serve as a potential target in the diagnosis and/or treatment of BPH.

## Figures and Tables

**Figure 1 biomedicines-10-01981-f001:**
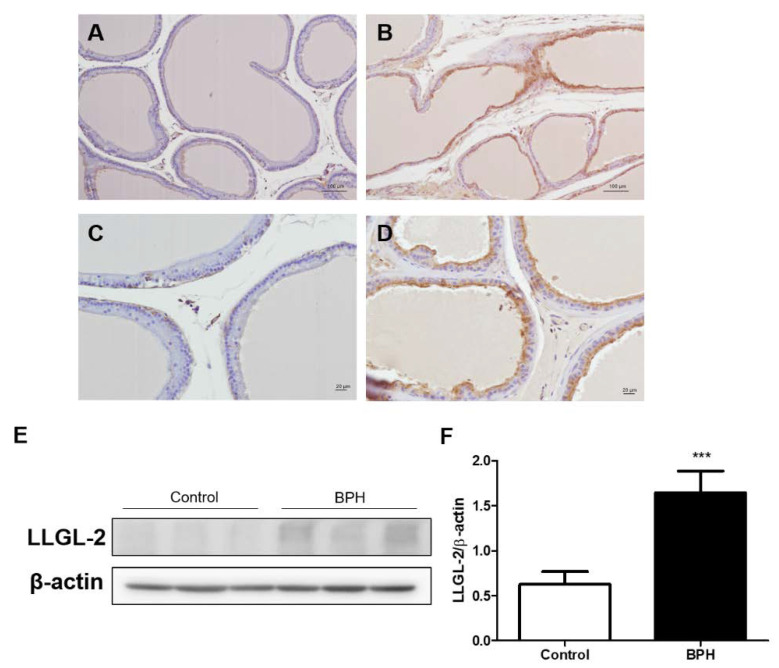
LLGL2 is highly expressed in testosterone-induced rat benign prostatic hyperplasia (BPH) in vivo. Representative photographs of immunostaining for LLGL2 (**A**–**D**). Prostate of control group; magnifications: ×100, scale bar = 100 μm (**A**) and magnifications: ×200, scale bar = 20 μm (**B**). Prostate of BPH group; magnifications: ×100, scale bar = 100 μm (**C**) and magnifications: ×200, scale bar = 20 μm (**D**). (**E**) LLGL2 protein expression in testosterone-induced rat BPH prostate tissue by Western blotting. (**F**) Graphical representation of the ratio of LLGL2 to β-actin. Values are expressed as the means ± standard deviation; *** *p* < 0.001 versus control group.

**Figure 2 biomedicines-10-01981-f002:**
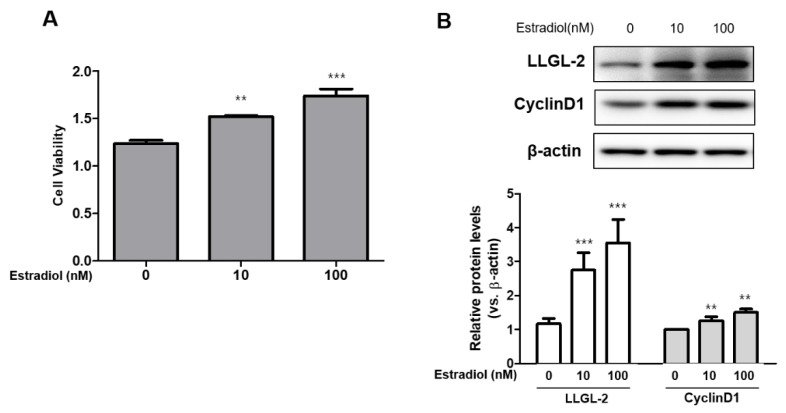
LLGL2 expression and BPH-1 cell proliferation in E2-treated BPH-1 cells. (**A**) E2 treatment (0, 10, and 100 nM) increased viability of BPH-1 cells. (**B**) Western blot data showing up-regulated protein expression of LLGL2 and cyclin D1 after treatment with E2 (0, 10, and 100 nM) for 24 h. Graphical representation of the ratio of LLGL2 and Cyclin D1 to β-actin. Values are expressed as the means ± standard deviation; ** *p* < 0.01 and *** *p* < 0.001 versus E0 group.

**Figure 3 biomedicines-10-01981-f003:**
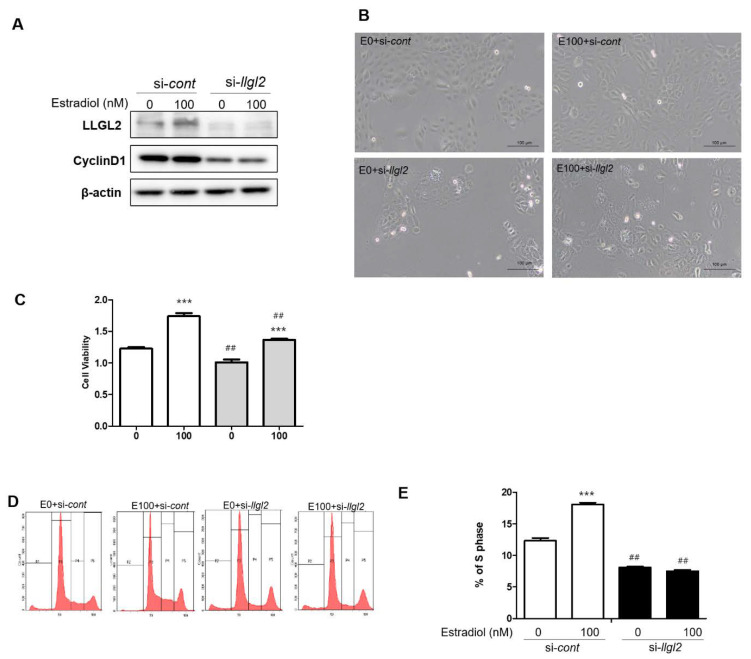
LLGL2 knockdown inhibits proliferation of E2-treated BPH-1 cells. BPH-1 cells were transiently transfected with a negative control siRNA (*si-cont*) or with LLGL2 siRNA (*si-llgl2*) against LLGL2 for 48 h. After that, cells were treated with E2 (0 and 100 nM) for 24 h. (**A**) Western blot data showing LLGL2 and Cyclin D1 protein expression levels in E2-treated BPH-1 cells after LLGL2 knockdown. (**B**) Cell morphology and confluence images of BPH-1 cells under a phase-contrast microscope (magnification 100×, scale bar = 100 μm). (**C**) Cell proliferation capacity of LLGL2 knockdown in E2-treated BPH-1 cells by CCK assay. (**D**) Cell cycle of LLGL2 knockdown in E2-treated BPH-1 cells detected by a flow cytometer. (**E**) Graphical representation of S-phase in LLGL2 knockdown in E2-treated BPH-1 cells. Values are expressed as the means ± standard deviation; *** *p* < 0.001 versus E0 + *si-cont* group. ## *p* < 0.01 versus E100 + *si-cont* group.

**Figure 4 biomedicines-10-01981-f004:**
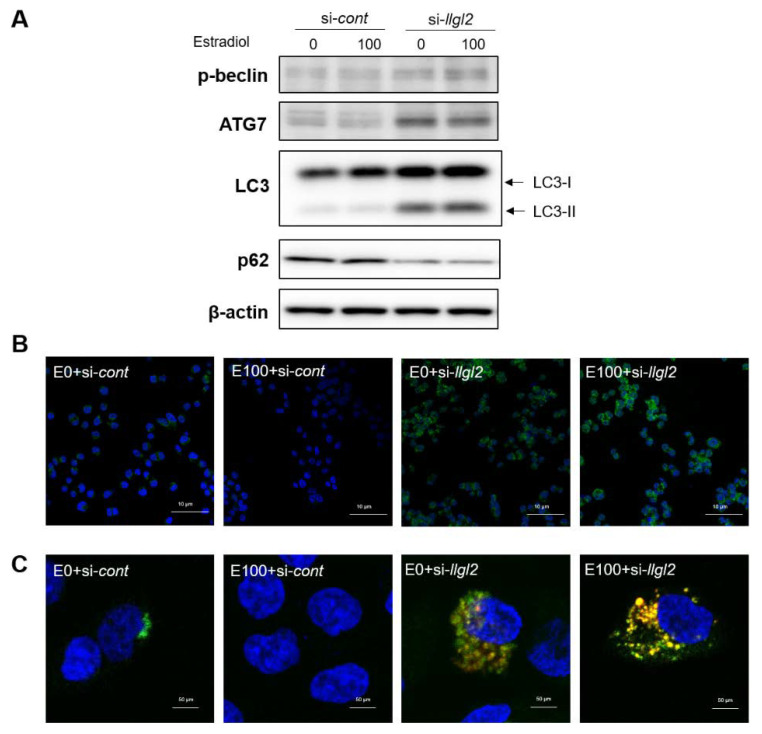
LLGL2 knockdown stimulates autophagosome formation in E2-treated BPH-1 cells. BPH-1 cells were transiently transfected with a negative control siRNA (*si-cont*) or with LLGL2 siRNA (*si-llgl2*) against LLGL2 for 48 h. After that, cells were treated with E2 (0 and 100 nM) for 24 h. (**A**) Western blot data of p-beclin, ATG7, LC3, and p62 protein expression levels in E2-treated BPH-1 cells after LLGL2 knockdown. (**B**) Representative images of endogenous LC3 puncta in E2-treated BPH-1 cells after LLGL2 knockdown. Cell nuclei were visualized by 6-diamino-2-phenylindole (DAPI; blue) staining and LC3-II was visualized with Alexa 488 conjugate (green) at 400× magnification. Scale bar = 10 μm. (**C**) Representative images of RFP-GFP-LC3 tandem fluorescent-tagged LC3 (RFP-GFP-LC3) in E2-treated BPH-1 cells after LLGL2 knockdown at 630× magnification. Scale bar = 50 μm.

**Figure 5 biomedicines-10-01981-f005:**
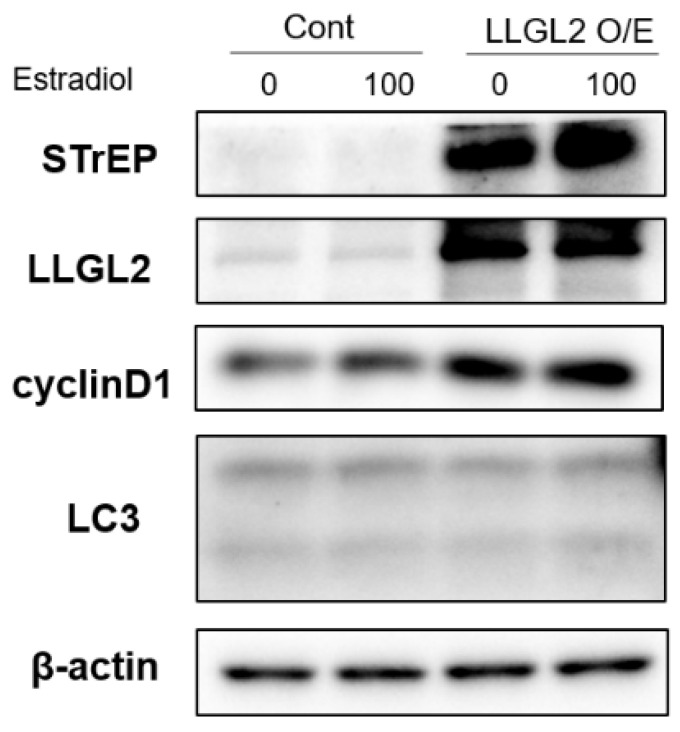
LLGL2 overexpression in E2-treated BPH-1 cells. BPH-1 cells were transiently transfected with a negative control vector (Cont) or with LLGL2 overexpression plasmid (LLGL2 O/E, Strep-tag pEXPR-IBA 105-LLGL2) for 24 h. After that, cells were treated with E2 (0 and 100 nM) for 24 h. Western blot data of STrEP, LLGL2, cyclinD1, and LC3 in E2-treated BPH-1 cells after LLGL2 overexpression are shown.

## Data Availability

The datasets generated and analyzed during the current study are available from the corresponding authors request.

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
