# Peer review of "Silencing of LLGL2 Suppresses the Estradiol-Induced BPH-1 Cell Proliferation through the Regulation of Autophagy"

_biomedicines, 2022, doi:10.3390/biomedicines10081981_

Round 1
Reviewer 1 Report
The Authors explore the role of LLGL2 in the prostate and suggest its role as a target in diagnosing and treating human BPH.
The manuscript is of interest, but, in my opinion, it requires major revision.
major concerns:
- In the results section 3.1, the sentence "To investigate the expression of LLGL2 in BPH in vivo, we used normal rat prostate and testosterone-induced rat prostate using IHC and western blotting. In testosterone-induced BPH tissue, LLGL2 has overexpressed especially the epithelial region of the prostate (Fig. 1B and 1D)." should be rephrased; it is unclear.
- Figure 1 should be improved using better-representing and more comparable images. It seems that each image has a different magnification. The scale bars in the images are missing.
- In the results section 3.2, the sentence "During the progression of BPH, the hormone is a pivotal regulatory factor for the proliferation of the prostate. ... Recently, E2 also affected the proliferation of prostatic cell proliferation [17]." should be rephrased; it is unclear.
- Figure 2 B should be improved using a better-representing panel since the expression of LLGL-2 is challenging to evaluate.
- Section 4. Discussion should be improved, including a more detailed discussion of the autophagy flux in respect of the results described in the manuscript. Indeed, the Authors stated that "silencing LLGL2 may suppress the proliferation of BPH-1 cells and promote autophagy." and "Overall, we identified the role of LLGL2 in the proliferation of the prostate which regulates the autophagosome formation." without clarifying or suggesting if there is an increment or a decrement of autophagic flux.
minor revision:
In the material and methods sections 2.5 and 2.6, the Authors should indicate more details, including which type of control they have performed.
In the materials and methods section 2.8, the Authors should be more precise about the secondary antibody they have used, "Alexa 488-conjugated secondary antibodies (Abcam)." Maybe they mean "Alexa Fluor" 488-...
In the materials and methods, sections 2.8 and 2.9
- the description of the use of confocal microscopy is not clear;
- A description of how the images were elaborate is missing, including which type of software they use;
The figure 3F should be probably the 3E
Author Response
Reviewer(s)' Comments to Author:
Reviewer: 1
Answer to Reviewer: 1
We are grateful to reviewer 1 for the critical comments and useful suggestions that have helped us to improve our paper considerably. As indicated in the following responses, we have incorporated all of these comments into the revised version of our paper.
The Authors explore the role of LLGL2 in the prostate and suggest its role as a target in diagnosing and treating human BPH.
The manuscript is of interest, but, in my opinion, it requires major revision.
major concerns:
- In the results section 3.1, the sentence "To investigate the expression of LLGL2 in BPH in vivo, we used normal rat prostate and testosterone-induced rat prostate using IHC and western blotting. In testosterone-induced BPH tissue, LLGL2 has overexpressed especially the epithelial region of the prostate (Fig. 1B and 1D)." should be rephrased; it is unclear.
Answer:
Thank you for your accurate and significant comments. As your suggestion, we rephrased this sentence. (rephrased part is red highlighted in the results)
[Rephrased part in Results]
To investigate the expression of LLGL2 in BPH in vivo, we used normal rat prostate and testosterone-induced rat prostate for IHC and western blotting. In testosterone-induced BPH tissues, LLGL2 was especially overexpressed in the epithelial region of the prostate (Figure 1B and 1D).
- Figure 1 should be improved using better-representing and more comparable images. It seems that each image has a different magnification. The scale bars in the images are missing.
Answer:
Thank you for your accurate and significant comments. As your suggestion, we revised images for figure 1. Upper panel (A and B) is the same magnification (x 100). The reason for panel A and panel B look slightly different magnification since B is a testosterone-induced BPH group that occurs hyperplasia and hypertrophy in the epithelial and stromal areas, so please understand it. Additionally, we added scale bars in the images and improved the image quality as below.
[Revised Figure 1]
- In the results section 3.2, the sentence "During the progression of BPH, the hormone is a pivotal regulatory factor for the proliferation of the prostate. ... Recently, E2 also affected the proliferation of prostatic cell proliferation [17]." should be rephrased; it is unclear.
Answer:
Thank you for your accurate and significant comments. As your suggestion, we rephrased this sentence. (rephrased part is red highlighted in the results)
[Rephrased part in Results]
During the progression of BPH, hormone is a pivotal regulatory factor for the proliferation of the prostate. Testosterone is one of the most potent hormones for prostatic proliferation. Recently, it has been shown that E2 can also affect prostatic cell proliferation
- Figure 2 B should be improved using a better-representing panel since the expression of LLGL-2 is challenging to evaluate.
Answer:
Thank you for your accurate and significant comments. As your suggestion, we performed western blotting again for the better results for LLGL2 and Cyclin D1 in Figure 2 as below.
[Revised Figure 2]
- Section 4. Discussion should be improved, including a more detailed discussion of the autophagy flux in respect of the results described in the manuscript. Indeed, the Authors stated that "silencing LLGL2 may suppress the proliferation of BPH-1 cells and promote autophagy." and "Overall, we identified the role of LLGL2 in the proliferation of the prostate which regulates the autophagosome formation." without clarifying or suggesting if there is an increment or a decrement of autophagic flux.
Answer:
Thank you for your accurate and significant comments. As your suggestion, we revised discussion part. (revised part is red highlighted in Discussion)
[Revised part in Discussion]
In this study, silencing LLGL2 expression upon E2 treatment showed enhanced autophagy through upregulation of LC3-II, ATG-7, and p-beclin and downregulation of p62. LC3-II, ATG-7, and p-beclin, and p62 known to be key regulatory molecules for autophagosome formation [24]. These results suggest that silencing of LLGL2 expression might regulate the autophagosome formation in E2-treated BPH-1 cells. Moreover, overexpression of LLGL2 was able to further prove these findings by showing the opposite results from the knockdown of LLGL2 in E2-treated BPH-1 cells. The present study revealed that silencing LLGL2 could suppress the proliferation of BPH-1 cells.
Overall, we suggested the role of LLGL2 in the proliferation of the prostate by the regulation of autophagy processes. This is the first attempt to study the role of LLGL2 in the prostate. These results provide a better understanding of the mechanisms involved in the role of LLGL2 in prostate cell proliferation. LLGL2 might serve as a potential target in the diagnosis and/or treatment of BPH.
minor revision:
In the material and methods sections 2.5 and 2.6, the Authors should indicate more details, including which type of control they have performed.
Answer:
Thank you for your accurate and significant comments. As your suggestion, we added more details for section 2.5 and 2.6 in materials and methods as below. (added part is red highlighted in materials and methods)
[Added part in section 2.5 and 2.6]
2.5. Observation of cell confluency
BPH-1 cells were seeded in a 6-well plate at a density of 1 × 105 cells/well and incubated in a 5% CO2 atmosphere at 37ËšC for 24 h. BPH-1 cells were treated with LLGL2 siRNAs for 48 h and were treated with E2 (0 and 100 nM) for 24 h. Cell morphology was observed with a phase-contrast microscope (Olympus, Japan). Photographs were taken at a magnification of 100 ×.
2.6. Flow cytometry
BPH-1 cells were seeded in a 6-well plate at a density of 1 × 105 cells/well and incubated in a 5% CO2 atmosphere at 37ËšC for 24 h. BPH-1 cells were treated with LLGL2 siRNAs for 48 h and were treated with E2 (0 and 100 nM) for 24 h. Cells were washed with PBS and were fixed with ice-cooled 70% ethanol at 4ËšC for 1 h. BPH-1 cells were stained with propidium iodide (PI, Invitrogen) and analyzed by flow cytometry (BD Bioscience, FranklinLakes, NJ, USA). Cell proliferation was expressed as the percentage of S-phase cells.
In the materials and methods section 2.8, the Authors should be more precise about the secondary antibody they have used, "Alexa 488-conjugated secondary antibodies (Abcam)." Maybe they mean "Alexa Fluor" 488-...
In the materials and methods, sections 2.8 and 2.9 the description of the use of confocal microscopy is not clear; A description of how the images were elaborate is missing, including which type of software they use;
Answer:
Thank you for your accurate and significant comments. As your suggestion, we revised the information of secondary antibodies and information of confocal microscopy as below. (revised part is red highlighted in materials and methods)
[Revised part in section 2.8 and 2.9]
2.8. Immunofluorescence
BPH-1 cells were fixed in 4% paraformaldehyde for 15 min, washed with PBS three times, and blocked with 3% bovine serum albumin for 30 min. These cells were then incubated with anti-LC3 (1:500, Sigma-Aldrich) at 4ËšC overnight. Cells were washed with PBS and incubated with Goat Anti-Rabbit IgG (Alexa Fluor® 488) conjugated secondary antibodies (Abcam) at RT for 1 h. The nucleus was counterstained with 4’6-diamino-2-phenylinodole (DAPI, Vector Labs). Zeiss LSM 880 with Airyscan confocal microscopy (Carl Zeiss, Jena, Germany) was performed at 488 and 592 nm wavelengths. Images were captured with a ZEN software.
2.9. Detection of autophagy flux
The formation of autophagosome and autolysosome in BPH-1 cells was detected using a Premo™ Autophagy Tandem Sensor RFP-GFP-LC3B Kit (Thermo Fischer Scientific, Grand Island, NY, USA) following manufacturer's instructions. The RFP-GFP-LC3B sensor could detect LC3B positive, neutral pH autophagosomes in green fluorescence (GFP) and LC3B positive acidic pH autophagolysosme in red fluorescence (RFP). BPH-1 cells were grown on coverslips and incubated with 6 μL of BacMam Reagents containing RFP-GFP-LC3B overnight. These cells were then treated with LLGL2 siRNAs alone for 48 h. The cells were washed with PBS three times. Nucleus was counterstained with DAPI (Vector Labs). Zeiss LSM 880 with Airyscan confocal microscopy (Zeiss) was performed at wavelengths of 488 and 592 nm Images were captured with a ZEN software.
The figure 3F should be probably the 3E
Answer:
Thank you for your accurate and significant comments. As your suggestion, we edited the figure number from 3F to 3E.

Reviewer 2 Report
The paper is interesting.
The language needs improvement.
Example the sentence “has overexpressed especially epithelial region of the prostate” could be replaced by “was overexpressed, especially in the epithelial region of the prostate”.
The entire first paragraph of the introduction needs to improve dramatically, both in syntax and in content. The link to estradiol is left incomplete.
Typing errors (Biomax, Seoul, Koreabyith the) need correction.
What can be done to improve the consistency of the protein gel runs for LLGL2? Some of the western blots seem to give fuzzy bands. Which parameter (protease inhibitors, RIPA, reagent freshness, timing, other?), is different from the gels that show a clearer band?
In figure 3E, some of the labels are too small to read.
The results that show silencing of LLGL2 expression to inhibit cell proliferation and to induce autophagosome formation in E2-treated BPH cells, while overexpression of LLGL2 to stimulate the proliferation of E2-treated BPH cells, are consistent with the manuscript title.
Is it possible to provide a final test of the autophagy effect by inhibiting autophagy in LLGL2-silenced cells? This could have practical and theoretical implications, and could amplify the impact of the article.
Author Response
Reviewer 2
Answer to Reviewer: 2
We are grateful to reviewer 2 for the critical comments and useful suggestions that have helped us to improve our paper considerably. As indicated in the following responses, we have incorporated all of these comments into the revised version of our paper.
The paper is interesting.
- The language needs improvement.
Example the sentence “has overexpressed especially epithelial region of the prostate” could be replaced by “was overexpressed, especially in the epithelial region of the prostate”.
The entire first paragraph of the introduction needs to improve dramatically, both in syntax and in content. The link to estradiol is left incomplete.
Typing errors (Biomax, Seoul, Koreabyith the) need correction.
Answer:
Thank you for your accurate and significant comments. As your suggestion, we re-edited English proof reading and revised typing errors. (English proofreading and revised typing errors in blue highlighted whole manuscript)
- What can be done to improve the consistency of the protein gel runs for LLGL2? Some of the western blots seem to give fuzzy bands. Which parameter (protease inhibitors, RIPA, reagent freshness, timing, other?), is different from the gels that show a clearer band?
Answer:
Thank you for your accurate and significant comments. As your suggestion, we performed western blotting again for the better results for LLGL2 in Figure 2. Unfortunately, we did western blotting several times for prostate tissue (figure 3). The results for western blotting always detect fuzzy bands. These fuzzy band results seem to be due to the difference in protein isolation between tissue and cells, so please understand this situation.
[Revised Figure 2]
- In figure 3E, some of the labels are too small to read.
Answer:
Thank you for your accurate and significant comments. As your suggestion, we revised the labels for Figure 3E. Thank you for your kind suggestion.
[Revised Figure 3]
- The results that show silencing of LLGL2 expression to inhibit cell proliferation and to induce autophagosome formation in E2-treated BPH cells, while overexpression of LLGL2 to stimulate the proliferation of E2-treated BPH cells, are consistent with the manuscript title.Is it possible to provide a final test of the autophagy effect by inhibiting autophagy in LLGL2-silenced cells? This could have practical and theoretical implications, and could amplify the impact of the article.
Answer:
Thank you for your significant comments. We are truly sorry, but the revision period was short just 10 days, so we were able to perform a simple western blotting results (figure 2). However, we could not do autophagy experiment as you suggest because it was not a sufficient period to conduct an additional experiment. Please kindly understand this part.
Please accept our warmest thanks for your significant suggestions and comments again and thanks for you give us a chance to revise the paper. We are pleased to make any possible modifications next. May you have a good health, happiness, and outstanding success in your life.
Best regards,
Ju-Young Jung

Round 2
Reviewer 1 Report
Thank you to the Authors for the punctual replies. The Authors significantly improved the manuscript that, in my opinion, can be accepted in the present form.